# 'Immunity Passports' for SARS-CoV-2: an online experimental study of the impact of antibody test terminology on perceived risk and behaviour

Jo Waller [1], G James Rubin,[2] Henry W W Potts,[3] Abigail L Mottershaw,[4] Theresa M Marteau [5]

¹School of Cancer and Pharmaceutical Sciences, King's College London, London, UK
²Department of Psychological Medicine, King's College London, London, UK
³Centre for Health Informatics and Multiprofessional Education, UCL, London, UK
⁴Behavioural Insights Team, London, UK
⁵Behaviour and Health Research Unit, University of Cambridge, Cambridge, UK

**Correspondence to**
Dr Theresa M Marteau;
tm388@cam.ac.uk

## ABSTRACT

**Objective** To assess the impact of describing an antibody-positive test result using the terms Immunity and Passport or Certificate, alone or in combination, on perceived risk of becoming infected with severe acute respiratory syndrome coronavirus 2 (SARS-CoV-2) and protective behaviours.

**Design** 2×3 experimental design.

**Setting** Online.

**Participants** 1204 adults from a UK research panel.

**Intervention** Participants were randomised to receive one of six descriptions of an antibody test and results showing SARS-CoV-2 antibodies, differing in the terms describing the type of test (Immunity vs Antibody) and the test result (Passport vs Certificate vs Test).

**Main outcome measures** Primary outcome: proportion of participants perceiving no risk of infection with SARS-CoV-2 given an antibody-positive test result. Other outcomes include: intended changes to frequency of hand washing and physical distancing.

**Results** When using the term Immunity (vs Antibody), 19.1% of participants (95% CI 16.1% to 22.5%) (vs 9.8% (95% CI 7.5% to 12.4%)) perceived no risk of catching coronavirus given an antibody-positive test result (adjusted OR (AOR): 2.91 (95% CI 1.52 to 5.55)). Using the terms Passport or Certificate—as opposed to Test—had no significant effect (AOR: 1.24 (95% CI 0.62 to 2.48) and AOR: 0.96 (95% CI 0.47 to 1.99) respectively). There was no significant interaction between the effects of the test and result terminology. Across groups, perceiving no risk of infection was associated with an intention to wash hands less frequently (AOR: 2.32 (95% CI 1.25 to 4.28)); there was no significant association with intended avoidance of physical contact (AOR: 1.37 (95% CI 0.93 to 2.03)).

**Conclusions** Using the term Immunity (vs Antibody) to describe antibody tests for SARS-CoV-2 increases the proportion of people believing that an antibody-positive result means they have no risk of catching coronavirus in the future, a perception that may be associated with less frequent hand washing.

**Trial registration number** Open Science Framework: https://osf.io/tjwz8/files/

## INTRODUCTION

At the height of the first wave of the COVID-19 pandemic, about a third of the world's population is estimated to have been in lockdown, with all but essential workers largely confined to home.[1]

Without an effective treatment or vaccine, testing for infection combined with contact tracing and isolation will be central to effective strategies to ease populations out of lockdown while keeping the basic reproduction number $(R_0)$ below 1.[2]

Testing for antibodies to severe acute respiratory syndrome coronavirus 2 (SARS-CoV-2) is a possible complement to testing for active infection to identify those who have developed antibodies to the virus and so may be able to return to work and other activities without significantly increasing transmission rates.[3] These tests have been variously described in the media as Immunity Passports,[4 5] Immunity Certificates,[6 7] Immunity Cards[8] and Release Certificates.[9] Unfortunately, the use of these terms implies a certainty unmatched by current evidence about antibody tests.[10] But whether these terms actually encourage a misplaced sense of certainty even before

testing is widely available is unknown and the focus of the current study.

Uncertainties inherent in tests for antibodies to SARS-CoV-2 include the extent and duration of immunity conferred.[11] They also include the uncertainties inherent in any test regarding the proportion of those who would be correctly identified. This depends on the test performance—its sensitivity and specificity—as well as the population prevalence of the tested condition.[12] Given these uncertainties, those who receive a test result indicating the presence of antibodies will have a residual risk of becoming infected by SARS-CoV-2 in the future.

Understanding that there is this residual risk—although one that is difficult to quantify at present—will be important to minimise transmission that could arise from those receiving 'antibody positive' test results. If people testing positive perceive that they have no risk of becoming infected by the virus, they may ignore any future symptoms of infection and facilitate transmission if they fail to self-isolate appropriately. Such a perception may also overgeneralise to a belief that they are unable to transmit infection through contact with contaminated surfaces. Regardless of antibody status, all individuals can indirectly transmit the virus between surfaces by touch. Hand washing or sanitising therefore needs to remain frequent.

Evidence from other testing programmes suggests that interpreting a low-risk result to mean no risk can be reduced by verbal and numerical expressions of residual risk when presenting test results.[13 14] But even before testing programmes are in place, the terms commonly used to describe these tests—Immunity Passport or Certificates—may inadvertently be fuelling a misplaced sense of certainty about their results. It is unknown whether describing these tests as being for immunity—as opposed to antibodies—or their results as passports or certificates increases misunderstanding of the residual risk inherent in an antibody-positive test result and thereby reducing adherence to protective behaviours and increasing risk of transmission.[15]

This study was designed to test two hypotheses: describing a test indicating the presence of antibodies using the term Immunity (vs Antibody), and describing test results as Passports or Certificates (vs Test), increases the likelihood that those with this test result erroneously perceive they have no risk of becoming infected in the future with coronavirus.

## METHODS
The protocol was preregistered on the Open Science Framework https://osf.io/tjwz8/ *Study 2*.

The statistical analysis plan was prespecified and uploaded to the Open Science Framework prior to receipt of the data https://osf.io/tjwz8/ *Study 2*.

An initial study with similar methods was conducted https://osf.io/tjwz8/ *Study 1* but, due to an error, the intervention was not correctly programmed. This study is therefore not reported.

## Design
The study was an online experiment using a 2×3 factorial design, with participants randomised, with an equal allocation ratio, to one of six groups varying in the description of an antibody test and a result showing the presence of antibodies. These descriptions differed only in the term used for what was being tested (Immunity vs Antibody) and the term used for the test result (Passport vs Certificate vs Test).

## Participants
A quota sample of 1204 adults was recruited via Predictiv, the Behavioural Insights Team's online experimentation platform (https://www.bi.team/bi-ventures/predictiv/) comprising 500 000 adults in the UK. Quotas were based on age, gender and UK region to achieve a sample broadly representative of the UK population. Up to 1373 clicked on the link to enter the study of whom 1214 subsequently completed the study. Ten were excluded for failing to meet quality checks (eg, the unique participant identifier was missing, or the participant had entered the survey more than once). Participants were reimbursed in points (equivalent to £1) which could be redeemed in cash, gift vouchers or charitable donations. Participants did not know the topic of the study prior to participation. All participants had provided opt-in consent to take part in surveys when they signed up to the online panel. No identifiable data were collected as part of the survey. All data collected by Predictiv are processed in line with their published privacy policy (http://www.predictiv.co.uk/privacy-policy.html).

## Patient and public involvement
Due to the rapid nature of this research, the public was not involved in the development of the study.

## Power
The sample size was chosen pragmatically without reference to a specific power calculation. We fitted a full linear model with two levels for test type (immunity/antibody), three levels for result type (passport/certificate/test) and an interaction term. Conservatively, we then had an 80% chance of detecting, at a 5% significance level, an increase in the primary outcome measure from 50% in a baseline group to 64% in another group. The power calculation was done using http://powerandsamplesize.com/, taking 50% as the baseline to provide a conservative estimate of power.

## Randomisation
Participants were randomised to groups by random number generation. A random number between 1 and 6 was generated for every participant on entry to the study to determine which description they saw, with each of the six numbers corresponding to one of the six descriptions. As this is based on true randomness, the number

**Table 1** Demographic characteristics of participants by experimental group (n=1204)

| | | Immunity | | | Antibody | | |
|---|---|---|---|---|---|---|---|
| | **All** | **Passport** | **Certificate** | **Test** | **Passport** | **Certificate** | **Test** |
| **Characteristics** | **(n=1204)** | **(n=187)** | **(n=235)** | **(n=179)** | **(n=219)** | **(n=209)** | **(n=175)** |
| Gender, n (%) | | | | | | | |
| Female | 606 (50.3) | 86 (46.0) | 126 (53.6) | 94 (52.5) | 112 (51.1) | 98 (46.9) | 90 (51.4) |
| Male | 598 (49.7) | 101 (54.0) | 109 (46.4) | 85 (47.5) | 107 (48.9) | 111 (53.1) | 85 (48.6) |
| Age, median (IQR) | 36 (32) | 35 (34) | 36 (34) | 36 (31) | 38 (32) | 34 (27) | 36 (31) |
| Education, n (%) | | | | | | | |
| Below degree | 888 (73.8) | 139 (74.3) | 177 (75.3) | 126 (70.4) | 165 (75.3) | 150 (71.8) | 131 (74.9) |
| Degree or above | 291 (24.2) | 45 (24.1) | 52 (22.1) | 49 (27.4) | 49 (22.4) | 55 (26.3) | 41 (23.4) |
| Missing | 25 (2.1) | 3 (1.6) | 6 (2.6) | 4 (2.2) | 5 (2.3) | 4 (1.9) | 3 (1.7) |
| UK region, n (%) | | | | | | | |
| England—London | 160 (13.3) | 24 (12.8) | 33 (14.0) | 27 (15.1) | 25 (11.4) | 27 (12.9) | 24 (13.7) |
| England—Midlands | 189 (15.7) | 38 (20.3) | 34 (14.5) | 24 (13.4) | 30 (13.7) | 32 (15.3) | 31 (17.7) |
| England—South and East | 373 (31.0) | 48 (25.7) | 74 (31.5) | 51 (28.5) | 75 (34.2) | 74 (35.4) | 51 (29.1) |
| England—North | 308 (25.6) | 46 (24.6) | 65 (27.7) | 50 (27.9) | 59 (26.9) | 43 (20.6) | 45 (25.7) |
| Scotland/Wales/NI | 174 (14.5) | 31 (16.6) | 29 (12.3) | 27 (15.1) | 30 (13.7) | 33 (15.8) | 24 (13.7) |

of participants within each group can vary due to chance (see table 1 for numbers in each group).

### Intervention

The intervention comprised a description of antibody testing and test results indicating the presence of antibodies (see box 1 for one example and (online supplementary material S1) for wording of all six descriptions). These differed across six groups in test name of results indicating the presence of antibodies. All descriptions included the information that the result would mean a lower risk of future infection and transmission, and that people with this result could return to work earlier.

### Outcome measures

Wording of the items used for each measure is shown in online supplementary material S2. Response options to all outcome measures are shown in table 2.

### Primary outcome

Proportion of participants perceiving an antibody-positive test result to mean no risk of catching coronavirus in

the future: assessed in response to a question with four response options.

### Secondary outcomes

Perceived likelihood of catching coronavirus in the future, assessed on a visual analogue scale from 0% to 100%.

Intention to engage in hand washing less or more frequently than now, given an antibody-positive test result: assessed in response to a question with five response options.

Intention to avoid physical contact with others outside the home more or less frequently than now, given an antibody-positive test result: assessed in response to a question with five response options.

Interest in undergoing the test if offered today: assessed in response to a question with four response options.

### Other measures

Demographic characteristics: age, gender, level of education and geographical region of residence. Employment status, planned to be included, was omitted due to a technical error.

---

**Box 1** **Immunity Passport: one of the six descriptions of an antibody-positive test result**

**Immunity Passport**

Scientists are developing tests to see who has already had coronavirus. No test is 100% effective.

This means that those who test 'positive' would have:

► Lower risk of catching coronavirus in the future—and therefore also·

► Lower risk of passing it on to others.

Those who test 'positive' would get an immunity passport.

They could return to work early.

---

### Statistical analyses

A detailed statistical analysis plan is available on the Open Science Framework, specified prior to receipt of the data https://osf.io/tjwz8/ *Study 2*. Binary logistic regression was used to assess the impact of test type (immunity/ antibody) and result type (passport/certificate/test) on the odds of believing the antibody test result means there is no risk of future infection. An interaction term was included in the model.[16] The analysis was repeated adjusting for age (including a quadratic function to model a non-linear relationship), gender, education and

**Table 2** Primary and secondary outcomes by experimental group (n=1204)

| | | Immunity | | | Antibody | | |
|---|---|---|---|---|---|---|---|
| | **All** | **Passport** | **Certificate** | **Test** | **Passport** | **Certificate** | **Test** |
| | **(n=1204)** | **(n=187)** | **(n=235)** | **(n=179)** | **(n=219)** | **(n=209)** | **(n=175)** |
| Perceived meaning of result for future risk, n (%) | | | | | | | |
| No risk | 174 (14.5) | 26 (13.9) | 50 (21.3) | 39 (21.8) | 24 (11.0) | 19 (9.1) | 16 (9.1) |
| Lower risk (in line with description) | 697 (57.9) | 106 (56.7) | 127 (54.0) | 92 (51.4) | 126 (57.5) | 134 (64.1) | 112 (64.0) |
| Average risk | 248 (20.6) | 40 (21.4) | 42 (17.9) | 36 (20.1) | 54 (24.7) | 41 (19.6) | 35 (20.0) |
| Higher risk | 85 (7.1) | 15 (8.0) | 16 (6.8) | 12 (6.7) | 15 (6.8) | 15 (7.2) | 12 (6.9) |
| Perceived absolute risk (0–100) (mean (SD)) | 37.7 (25.2) | 40.4 (26.5) | 37.0 (25.4) | 34.2 (27.1) | 39.5 (23.6) | 36.9 (25.2) | 37.9 (23.1) |
| Perceived residual risk, n (%) | | | | | | | |
| 1%–100% | 1144 (95.0) | 179 (95.7) | 219 (93.2) | 161 (89.9) | 214 (97.7) | 200 (95.7) | 171 (97.7) |
| 0% | 60 (5.0) | 8 (4.3) | 16 (6.8) | 18 (10.1) | 5 (2.3) | 9 (4.3) | 4 (2.3) |
| Intention to wash hands, n (%) | | | | | | | |
| Much less than now | 13 (1.1) | 3 (1.6) | 7 (3.0) | 0 (0.0) | 0 (0.0) | 2 (1.0) | 1 (0.6) |
| Less than now | 46 (3.8) | 6 (3.2) | 11 (4.7) | 7 (3.9) | 8 (3.7) | 10 (4.8) | 4 (2.3) |
| Same as now | 800 (66.4) | 121 (64.7) | 159 (67.7) | 127 (70.9) | 138 (63.0) | 143 (68.4) | 112 (64.0) |
| More than now | 161 (13.4) | 23 (12.3) | 26 (11.1) | 18 (10.1) | 34 (15.5) | 29 (13.9) | 31 (17.7) |
| Much more than now | 184 (15.3) | 34 (18.2) | 32 (13.6) | 27 (15.1) | 39 (17.8) | 25 (12.0) | 27 (15.4) |
| Intention to avoid physical contact, n (%) | | | | | | | |
| Much less than now | 36 (3.0) | 2 (1.1) | 9 (3.8) | 10 (5.6) | 4 (1.8) | 6 (2.9) | 5 (2.9) |
| Less than now | 201 (16.7) | 35 (18.7) | 48 (20.4) | 29 (16.2) | 32 (14.6) | 33 (15.8) | 24 (13.7) |
| Same as now | 642 (53.3) | 96 (51.3) | 116 (49.4) | 96 (53.6) | 116 (53.0) | 126 (60.3) | 92 (52.6) |
| More than now | 178 (14.8) | 28 (15.0) | 34 (14.5) | 21 (11.7) | 44 (20.1) | 27 (12.9) | 24 (13.7) |
| Much more than now | 147 (12.2) | 26 (13.9) | 28 (11.9) | 23 (12.8) | 23 (10.5) | 17 (8.1) | 30 (17.1) |
| Would you have the test if offered? n (%) | | | | | | | |
| No, definitely not | 38 (3.2) | 5 (2.7) | 14 (6.0) | 5 (2.8) | 5 (2.3) | 4 (1.9) | 5 (2.9) |
| No, probably not | 140 (11.6) | 24 (12.8) | 28 (11.9) | 23 (12.8) | 23 (10.5) | 20 (9.6) | 22 (12.6) |
| Yes, probably | 351 (29.2) | 54 (28.9) | 67 (28.5) | 53 (29.6) | 62 (28.3) | 58 (27.8) | 57 (32.6) |
| Yes, definitely | 675 (56.1) | 104 (55.6) | 126 (53.6) | 98 (54.7) | 129 (58.9) | 127 (60.8) | 91 (52.0) |

region based on prior results showing these are predictors of risk beliefs.

Binary logistic regressions were run (as above) for the secondary outcomes: intention to wash hands less, intention to avoid physical contact less and intention to undergo the test. Unadjusted and adjusted ORs (AOR) and 95% CIs are reported. Logistic regression was run to assess the extent to which intentions to engage in less frequent hand washing or social distancing measures is predicted by perceiving the test result to mean no risk of being infected in the future by coronavirus.

As only a very small proportion of participants gave a 'zero' response on the sliding scale of future risk, we used a linear regression model to examine this outcome, rather than a binary (zero vs other) logistic regression as prespecified in the analysis plan.

## Procedure

Data were collected using an online survey platform, Predictiv. On entry to the study, participants were informed that they were to be asked some questions about coronavirus and that it would take about 5 min to complete. Participants were then shown one of six brief descriptions of an antibody test for coronavirus (see online supplementary material S1 for full text for each of the six descriptions). They were then asked five questions, assessing the primary and secondary outcomes. Participants' demographic characteristics were accessed from the survey platform.

## RESULTS
### Sample characteristics
The sample comprised 606 women and 598 men with a median age of 36 years. Around a quarter had some

graduate-level education (24.2%) and there was good representation of all UK regions (see table 1). Distribution of sample characteristics by exposure group is shown in table 1.

### Descriptive statistics
#### Primary outcome
Responses to the five outcome questions for the whole sample and by experimental group are shown in table 2. Overall, 14.5% of respondents (95% CI 12.5% to 16.6%) interpreted the test result as meaning they had no risk of future infection. Over half (57.9% (95% CI 55.0% to 60.1%)) interpreted the test result as meaning that their future risk of coronavirus was 'lower', in line with the description they had been given.

#### Secondary outcomes
Perceived level of future risk (on a scale of 0%–100%) showed a complex, trimodal distribution. The median was 35% with an IQR from 18% to 51%. Only 5% of respondents put their risk at 0%. Overall, 63% put their risk below 50%. Ten per cent put their risk at 50%, which was the modal response. Twenty-four per cent put their risk at greater than 50%, but below 100%. Three per cent of respondents put their risk at 100%: that is, they said they were certain to contract the virus. There was a lack of consistency between responses to the categorical and the continuous measures of risk. Median responses on the sliding scale were 12.5%, 28.0%, 50.0% and 65.0% in the 'no risk', 'lower risk', 'average risk' and 'higher risk' categories, respectively, with responses ranging from 0 to 100 for all four of these groups. Although this suggests that a proportion of people were responding very differently to the two questions (eg, 'no risk' on the first outcome and

100% risk on the sliding scale), the increasing median responses with increasing risk category provide some reassurance that most people's responses were broadly consistent across the two measures.

On the behavioural outcomes, 4.9% (95% CI 3.8% to 6.3%) said they would wash their hands less frequently than now if they received a positive result while 19.7% (95% CI 17.5% to 22.1%) said they would be less inclined to avoid physical contact with others outside the home. Intentions to have the test if offered were high, with 56.1% (95% CI 53.2% to 58.9%) saying they would definitely, and 29.2% (95% CI 26.6% to 31.8%) saying they would probably have the test if offered today.

### Between-group differences in the primary outcome
When test type (Immunity vs Antibody), result type (Certificate vs Passport vs Test) and an interaction term were entered into a logistic regression model predicting the belief that the test result meant 'no risk' of future infection (see table 3), there was a significant effect of test type, which persisted when we adjusted for demographic factors (age (including a quadratic term), gender, education level and UK region; AOR: 2.91 (95% CI 1.52 to 5.55)). Those in the 'Immunity' group were more likely to believe the result meant 'no risk' than those in the Antibody group (19.1% (95% CI 16.1% to 22.5%) vs 9.8% (95% CI 7.5% to 12.4%)) (figure 1). There was no significant effect of result type and no significant interaction.

### Between-group differences in secondary outcomes
We analysed the continuous measures of future perceived risk of infection using a linear model (analysis of variance) with two levels for test type, three levels for result type and an interaction term. Overall, there was no significant effect:

**Table 3** Logistic regression analysis examining impact of test type and result type on perception that the test result means 'no risk'

| | Proportion answering 'no risk' in each subgroup % (95% CI) (n=1204) | Result means 'no risk' of future infection OR (95% CI) | |
| --- | --- | --- | --- |
| | | Mutually adjusted (n=1204) | Adjusted for demographics* (n=1179) |
| Test type | | | |
| Antibody (n=603) | 9.8 (7.5 to 12.4) | Ref | Ref |
| Immunity (n=601) | 19.1 (16.1 to 22.5) | **2.77 (1.48 to 5.17)** | **2.91 (1.52 to 5.55)** |
| Result type | | | |
| Test (n=354) | 15.5 (11.9 to 19.7) | Ref | Ref |
| Certificate (n=444) | 15.5 (12.3 to 19.3) | 0.99 (0.50 to 2.00) | 0.96 (0.47 to 1.99) |
| Passport (n=406) | 12.3 (9.3 to 15.9) | 1.22 (0.63 to 2.38) | 1.24 (0.62 to 2.48) |
| Test by result interaction | | | |
| Certificate by immunity result | | 0.98 (0.42 to 2.27) | 1.00 (0.42 to 2.40) |
| Passport by immunity result | | 0.47 (0.20 to 1.12) | 0.46 (0.19 to 1.12) |

Bold indicates statistical significance (p<.05)

*Fully adjusted model includes age (with quadratic term), gender, education level and region (coded as per table 1).

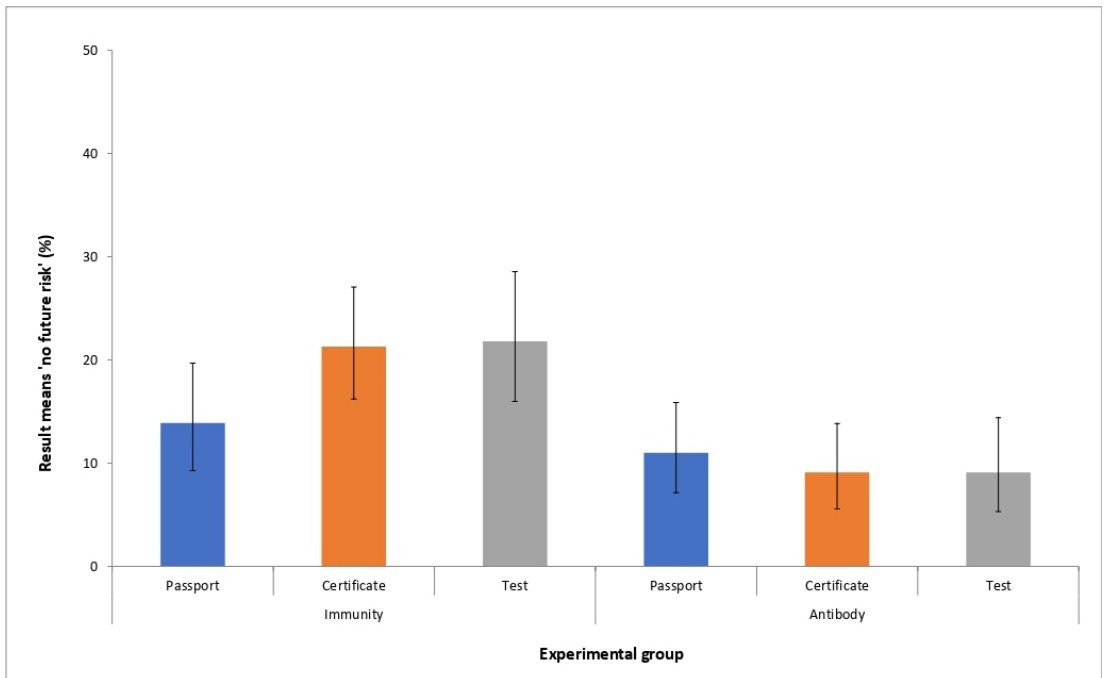

**Figure 1** Proportion believing an antibody-positive test result means 'no risk' of future infection. Error bars show 95% CIs.

$F_{5,1198}=1.46$, p=0.20, adjusted $R^2<1\%$. We repeated the analysis adjusting for demographic factors as covariates. Overall, there was a significant effect: $F_{13,1165}=1.88$, p=0.03, adjusted $R^2=1\%$. This was because of a significant effect of age: as age increased, perceived risk decreased. There remained no significant effect of the experimental variables.

Logistic regression analyses examining the impact of test type, result type and their interaction on intentions to wash hands and avoid physical contact less frequently and on willingness to have the test are shown in online supplementary tables 1 and 2. Neither test type, result type nor their interaction was significantly associated with these behavioural outcomes.

### Association between test result meaning 'no risk' and behavioural intentions

Logistic regression analyses were used to examine belief that the result meant 'no risk' as a predictor of intention to wash hands and avoid physical contact less frequently, given a positive test result (see figure 2 and online

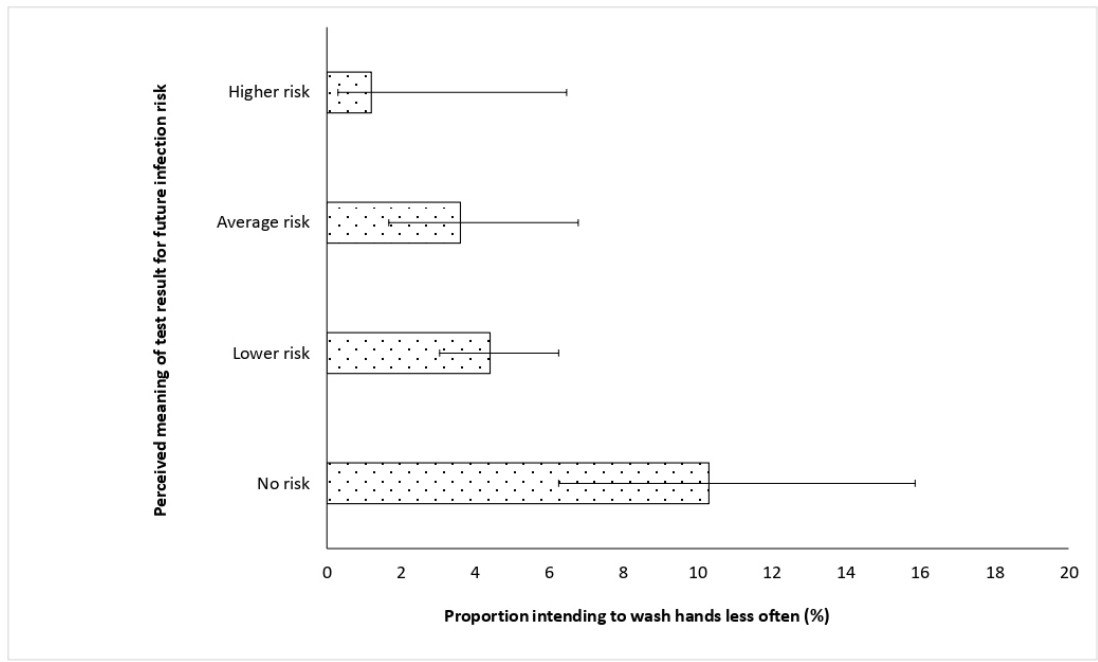

**Figure 2** Perceived meaning of an antibody-positive test result for future risk and intentions to reduce frequency of hand washing. Error bars show 95% CIs.

supplementary table 3). In analyses adjusting for demographic factors, those who believed there was no residual risk were at increased odds of intending to wash their hands less (AOR: 2.32 (95% CI 1.25 to 4.28)). The association with intentions to avoid physical contact was not significant (AOR: 1.37 (95% CI 0.93 to 2.03)).

## DISCUSSION

Using the term Immunity—as opposed to Antibody—to describe antibody tests for SARS-CoV-2 doubled the proportion who erroneously perceived they would have no risk of becoming infected with the virus in the future if they were given an antibody-positive test result, from 9.8% for Antibody to 19.1% for Immunity (AOR: 2.91 (95% CI 1.52 to 5.55)). Using the terms Passport, Certificate or Test to describe the results had no significant effect on risk perception (AOR: 0.96 (95% CI 0.47 to 1.99) for Certificate and 1.24 (95% CI 0.62 to 2.48) for Passport, compared with Test). The terms used to describe the test and results had no significant direct impact on intentions to engage in the protective behaviours of hand washing or physical distancing. However, across conditions, a greater proportion of those perceiving the result to mean no risk intended to wash their hands less often (10.3% (95% CI 6.3% to 15.9%)) compared with 4.0% who understood there was a residual risk (4.3% (95% CI 2.8% to 5.3%)).

There was no significant association with intended frequency of avoiding physical contact with others outside of the home. Interest in undergoing the test was high—with 85.2% saying they would probably or definitely have it if offered—and was unaffected by the terms used to describe the tests.

This study was designed to test two hypotheses, providing strong support for the first, that describing a test indicating the presence of antibodies using the term Immunity (vs Antibody) increases the likelihood that those with this test result erroneously perceive they have no risk of becoming infected in the future with coronavirus. This likely reflects a certainty about risk of future infection implicit in lay understandings of the term immunity that is not implied by the term antibody.[17] Qualitative studies could explore this and other potential mechanisms for the effect observed.

The results of this study did not support the second hypothesis that describing test results as Passports or Certificates increases the likelihood that those with this test result erroneously perceive they have no risk of becoming infected in the future with coronavirus. This does not mean that these terms are unproblematic however, only that they did not influence the specific perceptions that we explored. Qualitative studies are warranted to understand the broader meanings these terms have in the context of testing for antibodies for SARS-CoV-2 and other contexts.

Responses on the sliding scale of future risk showed a high variability and were largely unexplained by the experimental intervention or other variables measured.

They also showed only broad consistency with responses to the main outcome measure; some participants who believed the result meant 'no risk' then rated their future risk as 100%, and vice versa. This may point to considerable uncertainty in the public as to how to interpret test results and is consistent with the literature on risk communication and perception.[18 19] Some participants may have been responding solely on the information provided; others may have been using other information about coronavirus and immunity to make judgements about their own risk. It also likely reflects the well-described tendency of people to use a 50% response to indicate uncertainty rather than a true judgement of probability.[20] We also saw that about a quarter of respondents on the first question stated their risk was 'average' or 'higher'. Use of the top end of the scale is hard to interpret but may either reflect participants' not reading the information carefully and therefore misunderstanding the meaning of the result being communicated in the information, or their using information beyond the experiment to assess their risk. Participants likely considered many factors in making sense of the hypothetical result, such as their actual risk of contracting the virus based on their behaviour and likely exposure. Only 7% of the sample responded that their risk would be higher on the primary outcome measure. Reasons for this are unclear.

While we found no evidence for a direct effect on protective behaviours of the terms used to describe antibody test results, there was indirect evidence that perceiving no risk of future infection might reduce frequency of hand washing. This finding is tentative, given it is based on behavioural intentions in response to a hypothetical antibody-positive result. Nonetheless, the potential for antibody testing to increase viral transmission must be considered alongside the potential benefits the tests might have in allowing the easing of lockdown restrictions. Clear communication about the ongoing need for hand washing, in particular, will be essential and raising public awareness of the main mechanisms through which SARS-CoV-2 is transmitted—through air and surfaces—might help improve adherence. This, in addition to acknowledgement of the imperfect nature of the tests, will give the public a more accurate representation of the meaning and implications of an antibody test result and a better understanding of how to reduce the risk of transmission. Such communications need to emphasise that transmission can occur through contact regardless of antibody status. Such communications also need to be rigorously evaluated to ensure their effectiveness at communicating these points both to those undergoing antibody tests as well as to general populations that are now having to learn to live with SARS-CoV-2.

It was notable that 85% of respondents said that they would probably or definitely have an antibody test if they were offered it. This suggests high interest but should be treated with caution not least because fewer undergo than express interest in undergoing tests.[21]

## Strengths and limitations

This study provides the first experimental evidence for the potentially adverse impact on risk perceptions and protective behaviours of commonly used terms to describe SARS-CoV-2 antibody tests and their results. As such, it provides timely evidence to inform policy and research to mitigate these effects to realise the potential benefits of such tests.

The study has several limitations. First, participants were responding to a hypothetical test and asked to imagine that they had received a test result that had detected antibodies. The extent to which people responded using the information provided in the study is unknown. Some will likely have used other information about SARS-CoV-2 and their own experiences to answer the questions about risk. This is a limitation of all hypothetical scenario-based studies of this type but findings from such studies can generalise to clinical settings[22 23] although some caution is warranted. As antibody testing is not yet routinely being carried out in the UK, the impact of the wording used in the current study among those undergoing testing cannot be assessed. All scenarios used the term 'positive' so we were unable to examine any possible impact of the use of this term. In addition, due to the rapidity with which the current study was conducted, piloting of the materials in general population samples was not carried out. It is therefore possible that some of the information provided may have been misinterpreted.

Second, the protective behaviours of hand washing and physical distancing were measured using single items assessing behavioural intentions following a hypothetical test result.

Third, the sample size was insufficient to detect effect sizes that could be important at a population level. It is possible, for example, that the use of the terms Certificate or Passport might impact on risk perception, but the current study lacked the power to detect this.

Fourth, while quotas were used to achieve a sample broadly representative of the UK population, research panels are not representative of the general population.[24 25] We found no evidence that the impact of the interventions in this study was modified by demographic characteristics of the participants, providing some reassurance about the generalisability of results across age groups, gender, educational level and geographical region of the UK.

## Implications for research and policy

The results of this study have several implications for research and policy. Our finding that 85% of people were positively inclined towards having a test is encouraging for any future antibody testing programme. The effectiveness of antibody tests for SARS-CoV-2 will depend on high uptake, the extent and duration of any immunity conferred and the performance of a test, as well as on a good understanding of the meaning of test results among those who participate. First, the use of the term Immunity should be avoided in phrases to describe antibody tests, whether described as Passports, Certificates or Tests. This

has implications for the presentation of antibody testing by policymakers and the media, as well as those considering the wording of test result letters sent to tested individuals. Second, research is needed to evaluate different ways of informing those offered tests and receiving test results to minimise the proportion erroneously perceiving an antibody-positive test result to mean no risk of becoming infected with the virus. It should also focus on maximising understanding that—regardless of antibody status—anyone can indirectly transmit the virus by touching a contaminated surface and infecting the next surface they touch. Hand washing or sanitising therefore needs to remain frequent. Research is also needed with those undergoing actual tests, powered to detect effects judged meaningful in the context of a population-based testing programme and involving measures of actual behaviour.

## CONCLUSION

Interest in SARS-CoV-2 antibody testing is high—across many countries, employers and populations. While such testing could contribute to wider strategies to ease lockdown restrictions, their use may have an adverse impact on transmission-related behaviour. This appears to vary with the way the tests are described. Using the term Immunity (vs Antibody) to describe antibody tests increases the proportion of people believing that an antibody-positive result means they have no future risk of coronavirus, a perception that may be associated with less frequent hand washing and hence increased risk of transmission.

**Acknowledgements** We thank Steve Reicher for comments on an earlier draft of the study protocol.

**Contributors** The study was conceptualised by TM, JW and GJR. ALM completed the data collection. JW and HWWP analysed the data. All authors contributed to, and approved, the final manuscript.

**Funding** Data collection for this study was funded by a block UK government grant to the Behavioural Insights Team. JW is funded by a career development fellowship from Cancer Research UK (Ref C7492/A17219). GJR is funded by the National Institute for Health Research Health Protection Research Unit (NIHR HPRU) in Emergency Preparedness and Response at King's College London in partnership with Public Health England (PHE), in collaboration with the University of East Anglia.

**Disclaimer** The views expressed in this paper are those of the authors and not necessarily those of UK government, Cancer Research UK, NIHR or Public Health England.

**Competing interests** HWWP declares consultancy fees from Babylon Health.

**Patient and public involvement** Patients and/or the public were not involved in the design, or conduct, or reporting, or dissemination plans of this research.

**Patient consent for publication** Not required.

**Ethics approval** Ethical approval for this study was granted by the King's College London Research Ethics Committee (reference: MRA-19/20-18685).

**Provenance and peer review** Not commissioned; externally peer reviewed.

**Data availability statement** Data are available upon reasonable request. Anonymised data will be made available upon reasonable request.

and indication of whether changes were made. See: https://creativecommons.org/licenses/by/4.0/.

**ORCID iDs**

Jo Waller http://orcid.org/0000-0003-4025-9132

Theresa M Marteau http://orcid.org/0000-0003-3025-1129

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
