## [Reviewer comments · BMJ Open]

ARTICLE DETAILS

TITLE (PROVISIONAL)	"Immunity Passports" for SARS-CoV-2: an online experimental study of the impact of antibody test terminology on perceived risk and behaviour
AUTHORS	Waller, Jo; Rubin, GJ; Potts, Henry; Mottershaw, Abigail; Marteau, Theresa

VERSION 1 – REVIEW

REVIEWER	Jin Yong Lee Seoul National University Boramae Medical Center, Republic of Korea
REVIEW RETURNED	03-Jun-2020

GENERAL COMMENTS	This article is very interesting and appropriate study to BMJ Open readers in terms of how we describe the status of antibody positive to general public correctly and understandably. For better article, I would like to suggest some comments and recommendations. 1. I recommend authors to change the title " Appropriate terminology for antibody positive test result of SARS-CoV-2: an online experimental study in UK general public. Current title "Immunity Passport ~ " "Immunity" is better description versus "antibody", however, "passport" and "certificate" are no difference versus "test". Therefore, I would suggest the authors should change the title. 2. Regarding study design, I would have different idea. The authors designed 2*3 structure (Immunity vs Antibody; Passport, Certificate vs Test). However, if I designed this study, I would have compared (Immunity vs Antibody; Passport, Certificate vs Positive). Scientifically, antibody positive is correct terminology. Therefore, I believe the authors should have used the terminology "positive" instead of "test". If the authors used "test" terminology, respondents should have picked the following options; "Immunity Passport", "Immunity Certificate", "Immunity Positive", "Antibody Passport", "Antibody Certificate", "Antibody Positive". In this case, I think research results would have changed. Let me know why you used "test".
---

REVIEWER	Magda Osman Queen Mary University of London, UK
REVIEW RETURNED	05-Jun-2020

GENERAL COMMENTS	Clearly there is value to this study and it is quite clear that demonstrating that there are framing effects in the way test results are communicated is of interested. However, the study doesn't
--

	look like at the impact of 'what is being tested' x 'communication of test results' against the way in which actual test results from antibody tests is communicated to those that have access to being tested. This is a limitation unless the details of current serological tests can be matched in some to correspond with any of the variants included in the study. Presumably one reason that this wasn't done at the start is because the study was run before this knowledge would have been public? Anyway, to make this applicable to reality, there needs to be a careful attempt to examine the communication of current test results relative to hypothetical instances tested in the study. The above is a potentially easier concern to solve than the following one. There are multiple uncertainties that this project touches on, first we still don't know the actual infection prevalence rates, and we don't know the actual infection fatality rates - our uncertainty is constantly being adjusted to improving estimates over time by country and across countries. The accuracy of the serological tests with respect to false positive and false negative rates is also associated with some degree of uncertainty. The likelihood that having contracted the virus means that one has developed any immunity is controversial - especially given communication by the WHO. Why are all these details important? Because the instructions to all participants across all 6 manipulations includes "Scientists are developing tests to see who has already had coronavirus. No test is 100% effective. Directly followed by "This means that those who test 'positive' would have: lower risk of catching and therefore also lower risk of passing it on to others" The details here are misleading. Presumably the "This means" is supposed to directly follow from the first sentence not "No test is 100% effective" This seems to be a disclaimer that isn't placed in a way that makes any respondent see how it could directly be followed, meaningfully by "This means that...." because in the "This means that..." sentences that follow there isn't any implied uncertainty in what the test result would indicate, but that doesn't square with being told no test is 100% effective - which also is oddly phrased, since no test is 100% accurate - unless effective is meant to imply effective at detecting the presence of the virus. Also, the primary outcome of this study is assessing what estimate of risk people attribute to their susceptibility of the virus given a positive test result from a serological test, where the response options are no risk, lower risk (which appears in the instructions - so already primes people to that option - which is no doubt why across 6 conditions between 51% to 64% select it - being the most popular response) average risk, and higher risk. Where the authors identify that "lower risk" is the correct option. There is a 40 year history of work on risk communication which tells us that verbal expressions of probabilities leads to vast amounts of misinterpretation (unless they are accompanied by point estimates). For that reason, and taken together with the three critical sources of uncertainty i've outline (there are more), it isn't possible to be precise about what the correct inference is. Also, the phrasing of the primary outcome measure is ambiguous because for some of the options it says "Imagine you were given [an immunity passport/an immunity certificate/a result showing immunity/an antibody passport/an antibody certificate/a result showing antibodies]. But, again, let's leave that aside.
--	---

Let's say I read the instructions and then i pick the option "average risk". Is that an insane response? No. Why? Because, if I paid attention to the WHO's claims, and if I interpret average risk to mean same risk as the risk that anyone has if they haven't yet contracted the virus (or no change), then my inference is perfectly in line with a missive that says that people don't develop immunity to COVID-19 if they contract it.

Is this ultimately devastating for the study that has been run? No, but it requires a substantial rethink as to what can in fact be inferred by the responses to the primary outcome measure. What we don't know is what proportion select "low risk" because they are simply matching their response to what appeared in the instructions, what proportion select "low risk" because they do in fact think that having caught the virus they are less likely to catch it again (because immunity is temporary, or because the antibodies don't protect from other strains of the virus). The focus of the manuscript is on the "no risk" selected option, because this seems to most obviously discriminate between conditions ranging from ~9% to ~22%. But to my mind that isn't the only sub-group of interest, it just seems to be the slightly easier to interpret than the other groups.

For instance, why would any group say that the test increases their risk to the virus relative to before when they hadn't taken the test? This seems worth exploring. Also, given that people are then asked to estimate how likely it would be that they would contract the virus in the future the average response is ~40% with large degree of variance, so depending on what the modal response is, what is of interest is to what extent it matches up to the model response of the primary outcome measure. Finally, very little (actually not much at all) is made of what seems to be one of the most important findings of the entire piece which is the % of people that would definitely, probably, probably not, definitely not take the serological test. Across all 6 conditions ~ 16% (combining probably not, definitely not) would not take the test. That is an important figure to discuss. How does that compare to % of people that avoid taking tests for other highly infectious diseases? Is it very low or is it quite high? For a test that could make people's lives much easier and give them peace of mind, assuming the test is accurate to acceptable levels, then why is that across all 6 conditions the proportions of people no interested in taking it is low? Perhaps this was less salient an issue at the time the study was implemented, because now that conditions have changed then the focus and motivation of the sample might likely change. In sum, based on the points I've highlighted here, this study is interesting and informative but the data hasn't been examined to the depth that it deserves and the discussion doesn't consider many of the issues raised. These would need to be addressed to do justice to the study and the rich data set that has been gathered.

a few minor points

Page 7 line 4 - what was the error that meant that study 1 couldn't be reported?

Page 8 line 17 - what were the quality checks?

Page 8 - line 33 "We are fitting a full model with an interaction" - what model? You mean the design of the study as a 2 x3 - since g*power or equivalent wasn't used, I can't see what "model" means in this context. This needs clarifying. Also, references to the primary outcome measure are made without specifying what it is. Also I can't see how the values 64% were generated. Also,

	might be worth saying that a given the actual no. of participants that took part ~ 200 were allocated to each of the 6 conditions, or else specify the actual numbers per condition (or else point to Table 1 were the figures are presented). Page 9 - the responses options for the primary outcome need to be spelled out, this is essential information for the reader that needs to be in the text not referred to in Sup Mat. In fact it might be good to have a table with the critical questions and the response options that way all critical information is presented in the main body.
--	--

VERSION 1 – AUTHOR RESPONSE

Reviewers' Comments to Author:

Reviewer: 1

Reviewer Name: Jin Yong Lee

Institution and Country: Seoul National University Boramae Medical Center, Republic of Korea

Please state any competing interests or state 'None declared': None declared.

This article is very interesting and appropriate study to BMJ Open readers in terms of how we describe the status of antibody positive to general public correctly and understandably. For better article, I would like to suggest some comments and recommendations.

Thank you for this positive evaluation of the article.

1. I recommend authors to change the title "Appropriate terminology for antibody positive test result of SARS-CoV-2: an online experimental study in UK general public.

Current title "Immunity Passport ~ " "Immunity" is better description versus "antibody", however, "passport" and "certificate" are no difference versus "test". Therefore, I would suggest the authors should change the title.

Thank you for this suggestion. We have considered it carefully but feel that our title is more informative. The quotation marks around the term "Immunity Passport" are intended to make it clear that examining this terminology was the focus of the study without implying anything about the findings. We feel that using the term 'appropriate' in the title would be ambiguous, begging the question: appropriate for what?

2. Regarding study design, I would have different idea. The authors designed 2*3 structure (Immunity vs Antibody; Passport, Certificate vs Test). However, if I designed this study, I would have compared (Immunity vs Antibody; Passport, Certificate vs Positive). Scientifically, antibody positive is correct terminology. Therefore, I believe the authors should have used the terminology "positive" instead of "test". If the authors used "test" terminology, respondents should have picked the following options; "Immunity Passport", "Immunity Certificate", "Immunity Positive", "Antibody Passport", "Antibody Certificate", "Antibody Positive". In this case, I think research results would have changed. Let me know why you used "test".

Thank you for this comment. We used the term 'positive' in all the exposure conditions to describe the result, so we were not able to test its impact. 'Test' was also used in all the descriptions, so the condition we call 'Test' was really characterised by the absence of the terms 'passport' and 'certificate'. We agree that the findings may have been different had the word 'positive' been used in

the titles of Scenarios C and F. We have added an acknowledgment of this point in the Discussion on page 15.

'All scenarios used the term 'positive' so we were unable to examine any possible impact of the use of this term'.

Reviewer: 2

Reviewer Name: Magda Osman

Institution and Country: Queen Mary University of London, UK

Please state any competing interests or state 'None declared': none declared

Clearly there is value to this study and it is quite clear that demonstrating that there are framing effects in the way test results are communicated is of interested.

Thank you for this positive comment.

However, the study doesn't look like at the impact of 'what is being tested' x 'communication of test results' against the way in which actual test results from antibody tests is communicated to those that have access to being tested. This is a limitation unless the details of current serological tests can be matched in some to correspond with any of the variants included in the study. Presumably one reason that this wasn't done at the start is because the study was run before this knowledge would have been public? Anyway, to make this applicable to reality, there needs to be a careful attempt to examine the communication of current test results relative to hypothetical instances tested in the study.

Our aim was not so much to inform the wording of the delivery of results to the individual being tested, but rather to inform the way such results are discussed in the media and by policy-makers. As we describe in the Introduction, there has been frequent reference to 'immunity passports' in the media and by policy-makers and this broader discourse is likely to have at least as much impact on public perceptions and interpretations as the exact wording a results letter. Given that widespread antibody testing is still not available in the UK, we are unable to make direct comparisons. We have noted this in the Limitations section on page 15:

'As antibody testing is not yet routinely being carried out in the UK, the impact of the wording used in the current study amongst those undergoing testing cannot be assessed.'

The above is a potentially easier concern to solve than the following one. There are multiple uncertainties that this project touches on, first we still don't know the actual infection prevalence rates, and we don't know the actual infection fatality rates - our uncertainty is constantly being adjusted to improving estimates over time by country and across countries.

We agree that there are multiple uncertainties and that understanding of COVID-19 is evolving rapidly over time.

The accuracy of the serological tests with respect to false positive and false negative rates is also associated with some degree of uncertainty. The likelihood that having contracted the virus means that one has developed any immunity is controversial - especially given communication by the WHO. Why are all these details important? Because the instructions to all participants across all 6 manipulations includes "Scientists are developing tests to see who has already had coronavirus. No test is 100% effective. Directly followed by "This means that those who test 'positive' would have: lower risk of catching and therefore also lower risk of passing it on to others" The details here are misleading. Presumably the "This means" is supposed to directly follow from the first sentence not "No test is 100% effective"

Our intention here was that the statement about test efficacy not being 100% would explain why the risk was 'lower' (but not zero) among people testing positive. We agree that the logic may not be clear and have added a note to this effect in the Limitations section on page 15, linked to the fact that we were unable to carry out PPI due to the rapid nature of the research:

'In addition, due to the rapidity with which the current study was conducted, piloting of the materials in general population samples was not carried out. It is therefore possible that some of the information provided may have been misinterpreted.'

This seems to be a disclaimer that isn't placed in a way that makes any respondent see how it could directly be followed, meaningfully by "This means that...." because in the "This means that..." sentences that follow there isn't any implied uncertainty in what the test result would indicate, but that doesn't square with being told no test is 100% effective - which also is oddly phrased, since no test is 100% accurate - unless effective is meant to imply effective at detecting the presence of the virus.

We avoided the use of the term "accuracy" as this is a confusing term to use in the context of screening tests (see <https://medium.com/wintoncentre/live-facial-recognition-how-good-is-it-really-we-need-clarity-about-the-statistics-5140bd3c427d>). Instead we used the term effective to imply misclassification of antibody status by an antibody test and other uncertainties associated with antibody tests such as the level and duration of any immunity conferred.

Also, the primary outcome of this study is assessing what estimate of risk people attribute to their susceptibility of the virus given a positive test result from a serological test, where the response options are no risk, lower risk (which appears in the instructions - so already primes people to that option - which is no doubt why across 6 conditions between 51% to 64% select it - being the most popular response) average risk, and higher risk.

Our aim was to test the extent to which people would believe that the result conveyed 'lower risk' and attend to the fact that a residual risk remained, even if antibodies were detected.

Where the authors identify that "lower risk" is the correct option. There is a 40 year history of work on risk communication which tells us that verbal expressions of probabilities leads to vast amounts of misinterpretation (unless they are accompanied by point estimates). For that reason, and taken together with the three critical sources of uncertainty i've outline (there are more), it isn't possible to be precise about what the correct inference is.

We respectfully disagree with this analysis. If someone is told to imagine that they have a test result showing the presence of antibodies (and that their risk of future infection is therefore lower), we would argue that the 'lower' response is 'correct', if people have read and understood the information.

Also, the phrasing of the primary outcome measure is ambiguous because for some of the options it says "Imagine you were given [an immunity passport/an immunity certificate/a result showing immunity/an antibody passport/an antibody certificate/a result showing antibodies]. But, again, let's leave that aside.

We are unsure what is ambiguous here. In all the outcome questions, we were asking the respondent to imagine they had received the antibody positive test result and to answer accordingly.

Let's say I read the instructions and then i pick the option "average risk". Is that an insane response? No. Why? Because, if I paid attention to the WHO's claims, and if I interpret average risk to mean same risk as the risk that anyone has if they haven't yet contracted the virus (or no change), then my

inference is perfectly in line with a message that says that people don't develop immunity to COVID-19 if they contract it.

Again, given that the questions about perceived risk were all prefaced with a request to imagine receiving an antibody positive result, and a description of what this would mean, we were anticipating that the respondents would use that information to judge their risk, rather than thinking about their own experience or exposure to COVID-19. However, we acknowledge that the wide variety of responses, especially on the sliding scale outcome, suggest that other information was being used to make the risk judgement.

We had addressed some of these issues but have added some additional comments on page 14:

'Some participants may have been responding solely on the information provided; others may have been using other information about coronavirus and immunity to make judgements about their own risk.'

Is this ultimately devastating for the study that has been run? No, but it requires a substantial rethink as to what can in fact be inferred by the responses to the primary outcome measure. What we don't know is what proportion select "low risk" because they are simply matching their response to what appeared in the instructions, what proportion select "low risk" because they do in fact think that having caught the virus they are less likely to catch it again (because immunity is temporary, or because the antibodies don't protect from other strains of the virus).

We agree that we are unable to make this distinction and cannot assume that everyone was imagining receiving the test result, as instructed. We have added this as a limitation on page 15:

'The extent to which people responded using the information provided in the study is unknown. Some will likely have used other information about SARS-CoV-2 and their own experiences to answer the questions about risk. This is a limitation of all hypothetical scenario-based studies of this type.'

The focus of the manuscript is on the "no risk" selected option, because this seems to most obviously discriminate between conditions ranging from ~9% to ~22%. But to my mind that isn't the only subgroup of interest, it just seems to be the slightly easier to interpret than the other groups.

This is indeed the response we focus on, as specified in our pre-registered analysis plan. We did not choose to focus on this because it discriminated between conditions, but rather because we hypothesised this response might differ depending on the terminology used to describe the test result (with 'immunity' and 'passport' more easily interpreted to mean zero risk), and therefore built this into our analysis plan.

For instance, why would any group say that the test increases their risk to the virus relative to before when they hadn't taken the test? This seems worth exploring.

We agree this is an unexpected finding and we mention it at the bottom of page 14: 'Only 7% of the sample responded that their risk would be higher on the primary outcome measure. Reasons for this are unclear.'

Also, given that people are then asked to estimate how likely it would be that they would contract the virus in the future the average response is ~40% with large degree of variance, so depending on what the modal response is, what is of interest is to what extent it matches up to the modal response of the primary outcome measure.

This is a very good point. The relationship between responses to the two items is something we planned to explore and we have now added to our original description of responses to the sliding scale risk question to explain how this varied according to responses on the primary outcome measure (on page 8):

'There was a lack of consistency between responses to the categorical and the continuous measures of risk. Median responses on the sliding scale were 12.5%, 28.0%, 50.0% and 65.0% in the 'no risk', 'lower risk', 'average risk' and 'higher risk' categories respectively, with responses ranging from 0-100 for all four of these groups. Although this suggests that a proportion of people were responding very differently to the two questions (e.g. 'no risk' on the first outcome and 100% risk on the sliding scale), the increasing median responses with increasing risk category provides some reassurance that most people's responses were broadly consistent across the two measures'

And in the Discussion (page 14):

'They also showed only broad consistency with responses to the main outcome measure; some participants who believed the result meant 'no risk' then rated their future risk as

Finally, very little (actually not much at all) is made of what seems to be one of the most important findings of the entire piece which is the % of people that would definitely, probably, probably not, definitely not take the serological test. Across all 6 conditions ~ 16% (combining probably not, definitely not) would not take the test. That is an important figure to discuss. How does that compare to % of people that avoid taking tests for other highly infectious diseases? Is it very low or is it quite high? For a test that could make people's lives much easier and give them peace of mind, assuming the test is accurate to acceptable levels, then why is that across all 6 conditions the proportions of people not interested in taking it is low? Perhaps this was less salient an issue at the time the study was implemented, because now that conditions have changed then the focus and motivation of the sample might likely change.

Thank you for this suggestion. Our view is that the figure of those interested is high and have now added discussion of this on page 14:

'It was notable that 85% of respondents said that they would probably or definitely have an antibody test if they were offered it. This suggests high interest but should be treated with caution not least because fewer undergo than express interest in undergoing tests.'

In sum, based on the points I've highlighted here, this study is interesting and informative but the data hasn't been examined to the depth that it deserves and the discussion doesn't consider many of the issues raised. These would need to be addressed to do justice to the study and the rich data set that has been gathered.

Thank you. We hope that the revisions we have made in response to your comments have provided a richer discussion of the findings.

a few minor points

Page 7 line 4 - what was the error that meant that study 1 couldn't be reported?

An omission was made from the presentation of the scenarios so that the titles were not included. We felt the title (Immunity passport etc.) was important, and was a key aspect of the exposure.

Page 8 line 17 - what were the quality checks?

We have added this information on page 5:

'(e.g. the unique participant identifier was missing, or the participant had entered the survey more than once).'

Page 8 - line 33 "We are fitting a full model with an interaction" - what model? You mean the design of the study as a 2 x3 - since g*power or equivalent wasn't used, I can't see what "model" means in this context. This needs clarifying.

We have amended this line to make it clearer (page 6):

'We fitted a full linear model with two levels for test type (immunity/antibody), three levels for result type (passport/certificate/test) and an interaction term.'

Also, references to the primary outcome measure are made without specifying what it is.

The primary outcome is described in the Measures section on page 7:

'Proportion of participants perceiving an antibody-positive test result to mean no risk of catching coronavirus in the future, assessed in response to a question with four response options.'

Also I can't see how the values 64% were generated.

We have amended the description of the power calculation so it now reads (on page 6):

'The power calculation was done using <http://powerandsamplesize.com/>, taking 50% as the baseline to provide a conservative estimate of power.'

Also, might be worth saying that a given the actual no. of participants that took part ~ 200 were allocated to each of the 6 conditions, or else specify the actual numbers per condition (or else point to Table 1 were the figures are presented).

We have added a reference to Table 1 in the 'Randomisation' section on page 6.

Page 9 - the responses options for the primary outcome need to be spelled out, this is essential information for the reader that needs to be in the text not referred to in Sup Mat. In fact it might be good to have a table with the critical questions and the response options that way all critical information is presented in the main body.

The response options to all the items are shown in Table 2. We have added reference to Table 2 (as well as the supplementary material) in the Outcome Measures section on page 7.

VERSION 2 – REVIEW

REVIEWER	Magda Osman Queen Mary University of London
REVIEW RETURNED	22-Jun-2020

GENERAL COMMENTS	The authors include in the following response to the reviewers comments:
--

	Overall, I think the authors have responded to some of my comments with some degree of sensitivity, though not all. My comment: Where the authors identify that "lower risk" is the correct option. There is a 40year history of work on risk communication which tells us that verbal expressions of probabilities leads to vast amounts of misinterpretation (unless they are accompanied by point estimates). For that reason, and taken together with the three critical sources of uncertainty i've outline (there are more), it isn't possible to be precise about what the correct inference is. Authors' response: "We respectfully disagree with this analysis. If someone is told to imagine that they have a test result showing the presence of antibodies (and that their risk of future infection is therefore lower), we would argue that the 'lower' response is 'correct', if people have read and understood the information." This response reflects a misalignment with what the experimenters understanding, and what has been uncovered in a vast literature. The authors are making an assumption, and so their normative assumption is being used to characterise the appropriateness of the response as "correct", when the point of what was highlighted in my comment is that people will import extraneous information to inform their responses, and so they may well be valid in their responding based on their interpretation given what they are bringing to bear. So if the authors recast the responses to the key dependent measure according to most common and least comment, or else most aligned with the interpretation intended by the authors, then this is a better reflection, and more sensitive representation of the pattern of responses to the critical dependent measure. So, this needs to be caveated for all the reasons that I will not repeat that I outlined in a fairly long comment in my previous review. So, if the authors will state, a correct response in this task assumes that xxxxxx and so based on this interpretation, then the correct responses is xxx. However, we recognise that there are multiple interpretations because of the sources of information that people bring to bear to interpret the materials presented in our task... This isn't a minor point, because how the researchers report the responses to the dependent measure in the manuscript communicates a valued judgement about those responses. Someone that responds other than "I have a lower risk of catching coronavirus in the future" is deemed incorrect, but they are likely to have valid reason for their response. If this isn't recognised explicitly in the manuscript, then this is showing limited understanding of literature in judgement and decision-making, and huge literature on risk communication which has barely been referred to. Authors: "Our aim was not so much to inform the wording of the delivery of results to the individual being tested, but rather to inform the way such results are discussed in the media and by policy-makers. As we describe in the Introduction, there has been frequent reference to 'immunity passports' in the media and by policy-makers and this broader discourse is likely to have at least as much impact on public perceptions and interpretations as the exact wording a results letter. Given that widespread antibody
--	---

	testing is still not available in the UK, we are unable to make direct comparisons. We have noted this in the Limitations section on page 15:" I would suggest that this point needs to be highlighted at the start, in the introduction the comment mentioned above is not made explicit, and I think the manuscript would benefit it making this explicitly clear as well as in the limitations section, especially since this is something that appears in the highlights. On this point, what would be useful is to outline more precisely how the findings would inform policy, what policy? The recommendation focus specifically on how the details of antibody testing and implications of positive test results "...First, the use of the term Immunity should be avoided in phrases to describe antibody tests, whether described as Passports, Certificates or Tests. This has implications for the presentation of antibody testing by policy-makers and the media, as well as those considering the wording of test result letters sent to tested individuals". Presumably the findings regarding the % of people that would actually take the test is worth discussing as well.
--	---

VERSION 2 – AUTHOR RESPONSE

Reviewer Name: Magda Osman

Institution and Country: Queen Mary University of London, UK

Please state any competing interests or state 'None declared': none

The authors include in the following response to the reviewers comments:

Overall, I think the authors have responded to some of my comments with some degree of sensitivity, though not all.

My comment: Where the authors identify that "lower risk" is the correct option. There is a 40year history of work on risk communication which tells us that verbal expressions of probabilities leads to vast amounts of misinterpretation (unless they are accompanied by point estimates). For that reason, and taken together with the three critical sources of uncertainty i've outline (there are more), it isn't possible to be precise about what the correct inference is.

Authors' response: "We respectfully disagree with this analysis. If someone is told to imagine that they have a test result showing the presence of antibodies (and that their risk of future infection is therefore lower), we would argue that the 'lower' response is 'correct', if people have read and understood the information."

This response reflects a misalignment with what the experimenters understanding, and what has been uncovered in a vast literature. The authors are making an assumption, and so their normative

assumption is being used to characterise the appropriateness of the response as “correct”, when the point of what was highlighted in my comment is that people will import extraneous information to inform their responses, and so they may well be valid in their responding based on their interpretation given what they are bringing to bear. So if the authors recast the responses to the key dependent measure according to most common and least comment, or else most aligned with the interpretation intended by the authors, then this is a better reflection, and more sensitive representation of the pattern of responses to the critical dependent measure.

So, this needs to be caveated for all the reasons that I will not repeat that I outlined in a fairly long comment in my previous review. So, if the authors will state, a correct response in this task assumes that xxxxx and so based on this interpretation, then the correct responses is xxx. However, we recognise that there are multiple interpretations because of the sources of information that people bring to bear to interpret the materials presented in our task...

This isn't a minor point, because how the researchers report the responses to the dependent measure in the manuscript communicates a valued judgement about those responses. Someone that responds other than “I have a lower risk of catching coronavirus in the future” is deemed incorrect, but they are likely to have valid reason for their response. If this isn't recognised explicitly in the manuscript, then this is showing limited understanding of literature in judgement and decision-making, and huge literature on risk communication which has barely been referred to.

Response: We thank the reviewer for clarifying the focus of her concern. We can now see that the use of the term ‘correct’ is unhelpful and have amended ‘correctly interpreted the test result as meaning [...]’ to ‘interpreted the test result as meaning [...], in line with the description they had been given.’ We have also amended the wording in Table 2, to remove the term ‘correct’, replacing it with “as in the description”.

We agree that it would be useful for readers to place our study in the context of the broader literature on risk and communication, one that we are very familiar with, several of the authors having contributed to it. We have added to citations to two classic texts by leaders in the field – Paul Slovic and Barusch Fischhoff (page 15). We have also added further acknowledgement of the fact that participants were likely to have been using information beyond that provided in the study to make assessments about future risk:

‘Participants likely considered many factors in making sense of the hypothetical result, such as their actual risk of contracting the virus based on their behaviour and likely exposure.’ (p14, paragraph 2)

Authors: “Our aim was not so much to inform the wording of the delivery of results to the individual being tested, but rather to inform the way such results are discussed in the media and by policy-makers. As we describe in the Introduction, there has been frequent reference to ‘immunity passports’ in the media and by policy-makers and this broader discourse is likely to have at least as much impact on public perceptions and interpretations as the exact wording a results letter. Given that widespread antibody testing is still not available in the UK, we are unable to make direct comparisons. We have noted this in the Limitations section on page 15.”

I would suggest that this point needs to be highlighted at the start, in the introduction the comment mentioned above is not made explicit, and I think the manuscript would benefit it making this explicitly clear as well as in the limitations section, especially since this is something that appears in the highlights.

Response: Our intention was to inform the use of language in the wider discourse on antibody testing, not just the communication of results. We have clarified that the focus of the study was upon generating evidence regarding a concern that the use of terms such as Immunity Passports would lead to a misleading sense of certainty:

'But whether these terms actually encourage a misplaced sense of certainty even before testing is widely available is unknown and the focus of the current study.' (p4, paragraph 3).

On this point, what would be useful is to outline more precisely how the findings would inform policy, what policy? The recommendation focus specifically on how the details of antibody testing and implications of positive test results "...First, the use of the term Immunity should be avoided in phrases to describe antibody tests, whether described as Passports, Certificates or Tests. This has implications for the presentation of antibody testing by policy-makers and the media, as well as those considering the wording of test result letters sent to tested individuals". Presumably the findings regarding the % of people that would actually take the test is worth discussing as well.

Response: We have added a line on the policy implication of the finding that 85% of people said they would have testing:

'Our finding that 85% of people were positively inclined towards having a test is encouraging for any future antibody testing programme.' (page 15, final paragraph).